# Synthesis of Fucosyl-Oligosaccharides Using α-l-Fucosidase from *Lactobacillus rhamnosus* GG

**DOI:** 10.3390/molecules24132402

**Published:** 2019-06-29

**Authors:** Yolanda Escamilla-Lozano, Francisco Guzmán-Rodríguez, Sergio Alatorre-Santamaría, Mariano García-Garibay, Lorena Gómez-Ruiz, Gabriela Rodríguez-Serrano, Alma Cruz-Guerrero

**Affiliations:** 1Departamento de Biotecnología, Universidad Autónoma Metropolitana-Iztapalapa, Cd. de México 09340, Mexico; 2Departamento de Ciencias de la Alimentación, Universidad Autónoma Metropolitana-Lerma, Edo. México 52006, Mexico

**Keywords:** fucosidase, fucosyl-oligosaccharides, *Lactobacillus rhamnosus* GG, transfucosylation

## Abstract

Fucosyl-oligosaccharides are natural prebiotics that promote the growth of probiotics in human gut and stimulate the innate immune system. In this work, the release of α-lfucosidase by *Lactobacillus rhamnosus* GG, and the use of this enzyme for the synthesis of fucosyl-oligosaccharides were investigated. Since α-lfucosidase is a membrane-bound enzyme, its release from the cells was induced by addition of 4-nitrophenyl-α-l-fucopyranoside (*p*NP-Fuc). Enzyme activity associated with the cell was recovered at 78% of its total activity. Fucosyl-oligosaccharides where synthesized using α-l-fucosidase extract and *p*NP-Fuc as donor substrate, and D-lactose or D-lactulose as acceptor substrates, reaching a yield up to 25%. Fucosyllactose was obtained as a reaction product with D-lactose, and its composition was confirmed by mass spectrometry (MALDI-TOF MS). It is possible that the fucosyl-oligosaccharide synthesized in this study has biological functions similar to human milk oligosaccharides.

## 1. Introduction

Fucosyl-oligosaccharides are present on cell surfaces and in blood group antigens, intestinal mucin, and human milk. Their anti-adhesive activity toward infective microorganisms has been reported, thereby preventing adhesion of pathogens to infant mucosal surfaces and decreasing the risk for bacterial infections and diarrhea [1]. Furthermore, the most abundant human milk oligosaccharides (HMOs) are fucosyl-oligosaccharides, proven to inhibit adhesion of pathogenic microorganisms such as *Campylobacter jejuni*, *Vibrio cholerae*, *Salmonella*, and *Shigella*. These oligosaccharides are natural prebiotics and stimulate the innate immune system [2,3,4]. Therefore, the synthesis of fucosyl-oligosaccharides has currently been receiving great attention, due to their important roles in the many biological processes described above. 

There is great interest in enzymatic HMO synthesis, not only due to the fact that chemical synthesis involving saccharides is a cumbersome task, usually relying on multiple steps, complicated protective group manipulations, and the use of potentially toxic reagents [5]. Several examples of glycosidases applied in the synthesis of oligosaccharides have been reported in the literature [6,7]. Particularly, α-lfucosidases can be employed for the production of valuable compounds by transfucosylation reactions, which are crucial intermediates in the synthesis of HMOs.

α-lFucosidases play an important role in the metabolism of biological substrates containing L-fucose. These enzymes have been isolated from prokaryotic and eukaryotic cells and are classified into four groups: those from microorganisms [8,9], marine mollusks [10], plants [11], and mammals [12]. Their substrate specificity differs depending on their origin. 

There are few reports of α-lfucosidases from *Lactobacillus*, a genus that includes several probiotic bacteria which are common in the human intestine [13,14]. A study of the genomic analysis of 25 *Lactobacillus* species reveals that *Lactobacillus rhamnosus* encodes putative α-lfucosidases [15]. In particular, Escamilla-Lozano et al. [13] reported the synthesis of cell-associated α-lfucosidases by *Lactobacillus rhamnosus* GG employing different carbon sources. Rodríguez-Díaz et al. [16] employed the recombinant fucosidases AlfB and AlfC from *Lactobacillus casei* BL23 to synthesize fucosyl-α-(1-3)-*N*-acetyl-d-glucosamine and fucosyl-α-(1-6)-*N*-acetyl-d-glucosamine, respectively. Therefore, the aim of this work was to study the release of α-lfucosidase from *Lactobacillus rhamnosus* GG, and the use of this enzyme for the synthesis of fucosyl-oligosaccharides with composition similarity to those found in human milk.

## 2. Results and Discussion

### 2.1. α-l-Fucosidase Release

Figure 1 shows the α-lfucosidase released from cultured cells when exposed to 4-nitrophenyl-α-lfucopyranoside (pNP-Fuc), recovering 78% of its total activity. The obtained supernatant containing enzyme extract was set aside and stored for further use in the synthesis of fucosyl-oligosaccharides. Interestingly, the release of α-lfucosidase promoted by *p*NP-Fuc suggests that this compound should be playing an important role in the enzyme excretion by the cells. Sikkema et al. [17] indicated that apolar compounds, such as nitrophenol, could easily penetrate the lipid bilayer of the cytoplasmic membrane, potentially causing significant changes in its structure and integrity. Furthermore, accumulation of apolar compounds in the membrane can lead to modification of membrane fluidity which may manifest by swelling of the lipid bilayer. Additionally, the lipid annulus which surrounds membrane-embedded proteins would also be affected causing altered protein conformations. This could explain why membrane-bound α-lfucosidase was released into the medium when the microorganism was in the presence of *p*NP-Fuc. Moreover, it was observed that activity of α-lfucosidase was conserved in cell membrane debris (data not shown). This data would indicate that the *L. rhamnosus* GG α-lfucosidase is a membrane-bound enzyme, similar to what is observed for α-lfucosidase from *Bifidobacterium bifidum* [18]. Dong et al. [19] reported that after chromatographic steps, 27% of the intracellular α-l-fucosidase from Wenyingzhuangia fucanilytica was recovered, while Eneyskaya et al. [20] obtained a yield of 30% of the initial activity of the intracellular α-l-fucosidase from *Thermus* sp.

### 2.2. Synthesis of Fucosyl-Oligosaccharides 

Figure 2a shows a typical chromatogram of the products obtained during the transfucosylation reaction. The retention time registered for the synthesized fucosyl-oligosaccharide (15.7 min) was similar to that of 2′-fucosyllactose (15.6 min) (Figure 2b). As can be noted in Figure 2a, during the course of the enzymatic reaction, α-lfucosidase performed the hydrolysis of *p*NP-Fuc as well as transfucosylation, resulting in the production of fucosyl-oligosaccharide. 

Figure 3 shows the reaction course of a typical transfucosylation reaction when using D-lactose or D-lactulose as acceptor substrate. The final concentration of fucosyl-oligosaccharides synthesized after 12 h reaction was 1.16 µmol/mL for D-lactose and 0.87 µmol/mL for D-lactulose. The ability to synthesize fucosyl-oligosaccharides employing α-l-fucosidase from *L. rhamnosus* GG is reported here for the first time, opening a wide field of investigation into the transfucosylation reaction to obtain HMO mimics.

The enzymatic synthesis of fucosyl-oligosaccharides offers the advantage of forming specific glycosidic linkages in the presence of other reactive functional groups. Furthermore, transfucosylation reactions demand an activated donor substrate, which means that the fucose moiety that is to be transferred must form a high energy bond with a good leaving group, so that the energy provided by cleaving this bond contributes to lowering the activation energy in the transglycosylation reaction, and the most reported substrate in transfucosylation reactions is *p*NP-Fuc [7,21,22]. The transglycosylation activity of α-l-fucosidases is generally moderate compared with the hydrolysis activity, although it is variable and depends on the origin of the enzyme. Rodriguez-Diaz et al. [16] reported that α-l-fucosidases AlfB and AlfC from *L. casei* BL23 can able to synthesize by transglycosylation the disaccharides fucosyl-α-(1-3)-*N*-acetylglucosamine and fucosyl-α-(1-6)-*N*-acetylglucosamine, respectively. Zeng et al. [23] studied the regioselectivity of α-l-fucosidase from *Alcaligenes* sp. to fucosylate *p*-nitrophenyl glycosides (*p*NP-lactosamine, *p*NP-lactose) using *p*NP-Fuc as the donor. In both cases, fucosylation occurred preferably on carbon 3 of galactose. Zeuner et al. [7] studied the ability of the BbAfcB fucosidase from *B. bifidum* JCM 1254 to transfer fucose from 3-fucosyllactose to lacto-*N*-tetraose for the synthesis of lacto-*N*-fucopentaose II, and reported that this fucosidase is regioselective toward both hydrolysis and the formation of α-1,3/4 fucosidic bonds. We are currently studying the regioselectivity of the α-l-fucosidase of *L. rhamnosus* GG to reveal which isomers of the fucooligosaccharides can be obtained during the transfucosylation reaction.

As summarized in Table 1, transfucosylation using D-lactose or D-lactulose as acceptor substrate afforded the desired oligosaccharides at 21% and 25% yield, respectively. The yields are comparable with those reported previously by Ajisaka and Shirakabe [24] for fucosidase from *Corynebacterium* sp. when methyl-β-d-galactopyranoside (25%) and D-galactose (18%) were used as acceptor substrates. Likewise, Farkas et al. [25] reported yields of 25% to 29% using fucosidase from *Penicillium multicolor* with different acceptor substrates. Rodríguez-Díaz et al. [16] reported that α-lfucosidase AlfB from *L. casei* BL23 was able to synthesize fucosyl-α-(1-3)-*N*-acetylglucosamine with a yield of 23%, while the α-lfucosidase AlfC synthesized fucosyl-α-(1-6)-*N*-acetyl-d-glucosamine with a yield of 56%. Guzmán-Rodríguez et al. [26] employed α-lfucosidase from *Thermotoga maritima* for the synthesis of fucosyllactose with a of yield 32.5%. Furthermore, Zeuner et al. [6] and Petschacher and Nidetzky [27] reported that typically, wild-type fucosidases can reach a 30–40% yield in transfucosylation reactions.

On the other hand, when D-galactose was used as acceptor substrate, no transfer (Table 1) or hydrolysis reaction were observed, which was verified by the fact that 4-nitrophenol (*p*NP) was absent in the reaction media. Reglero and Cabezas [28], and Grove and Serif [29] reported the D-galactose-induced inhibition of fucosidases from boar and mollusk, giving rise to the hypothesis that α-lfucosidase from *L. rhamnosus* GG could also exhibit inhibitory effects.

### 2.3. Composition of Synthesized Fucosyl-Oligosaccharide

To prove that the synthesized compound was fucosyllactose, when using D-lactose as acceptor substrate, the synthesized compound was purified and subjected to acid hydrolysis. Figure 4a shows a chromatogram of the purified fucosyl-oligosaccharide. Using HPLC, the moieties released in equimolar concentration from acid-hydrolyzed compound were identified as D-glucose, D-galactose, and L-fucose, as shown in Figure 4b, supporting the formation of the fucosyl-oligosaccharide. 

Purified fucosyl-oligosaccharide was subjected to MALDI-TOF MS analysis. In the mass spectrum of the compound (Figure 5a), the strongest signal (*m*/*z* 511.008) corresponds to the mass of a fucosyllactose-Na^+^ adduct. Subsequent fragmentation of the molecular ion (Figure 5b) reveals the presence of three hexoses, where one is a fucose moiety. Data from the MS/MS spectrum in conjunction with acid hydrolysis/HPLC confirms that the synthetized oligosaccharide corresponds to fucosyllactose. Guzmán-Rodríguez et al. [26] employed MALDI-TOF MS to identify a fucosyllactose produced via transglycosylation by α-l-fucosidase from *Thermotoga maritima.* They found a fucose and two hexoses in the structure of the produced fucosyl-oligosaccharide. It has been reported that human milk fucosyl-oligosaccharides have shown an anti-infectious effect against several pathogens. Ruiz Palacios et al. [30] found that α-1,2-fucosyl-oligosaccharides inhibited the adhesion of *Campylobacter jejuni* to HEp-2 cell line. Weichert et al [31] reported that 2′-fucosyllactose inhibited the adhesion of *Salmonella enterica* serovar *fyris* to Caco-2 cell line. This last group also reported that 2′-fucosyllactose, as well as its isomer 3-fucosyllactose, inhibit the adhesion of enteropathogenic *Escherichia coli* and *Pseudomonas aeruginosa* to different human cell lines, showing that at least two isomers of the same fucosyl-oligosaccharide can express a similar bioactivity. In addition, Sotgiu et al. [32] reported that 2′-fucosyllactose has a positive effect on the immune system, since it increases the production of the anti-inflammatory cytokine IL-10, and decreases the levels of the proinflammatory IL-12 in mononuclear cells. Duska-McEwen et al. [33] reported that 2′-fucosyllactose and 3-fucosyllactose can structurally mimic histo-blood group antigens and block the binding of norovirus, which can cause acute gastroenteritis in humans.

## 3. Materials and Methods 

### 3.1. Materials

Yeast extract and casein peptone were purchased from B. D. Bioxon (Mexico City, Mexico). *p*NP-Fuc, *p*NP, D-lactose, D-glucose, D-galactose, L-fucose, and Man Rogosa and Sharpe (MRS) agar were purchased from Sigma-Aldrich (St. Louis, MO, USA), 2′-fucosyllactose was purchased from Carbosynth (Berkshire, UK). Sodium phosphate and sodium hydroxide were purchased from J. T. Baker (Mexico City, Mexico). Milli-Q® (Billerica, MA, USA) water was used throughout the experiments.

### 3.2. Microorganism

*Lactobacillus rhamnosus* GG, previously isolated by Cruz-Guerrero et al. [34], was used in this study. The culture was stored in MRS agar at 4 °C.

### 3.3. Production of α-lFucosidase

Production of α-lfucosidase was performed as previously described by Escamilla-Lozano et al. [13]. In brief, *L. rhamnosus* GG was cultivated in a culture medium at 37 °C for 24 h. The culture medium contained 5 g/L yeast extract, 10 g/L casein peptone, and 10 g/L D-glucose. Fermentation broth was centrifuged at 3110× *g* (Beckman J2-MI, Beckman Instruments, Palo Alto, CA, USA) for 20 min at 4 °C. Cellular pellet was resuspended in phosphate buffer (100 mM, pH 7), to analyze for α-l-fucosidase activity.

### 3.4. Release of α-lFucosidase

Cell pellets, obtained as described above, were washed twice with isotonic solution (NaCl 0.9%). The cells were added to a solution containing 0.5 g/L of *p*NP-Fuc and incubated at 37 °C for 48 h. Samples were taken periodically and centrifuged at 3110× *g* for 20 min at 4 °C. Both supernatant and cellular pellet were resuspended in phosphate buffer (100 mM, pH 7) and analyzed for α-lfucosidase activity. Released enzyme in the supernatant was used for the synthesis of fucosyl-oligosaccharides.

### 3.5. Enzyme Activity Assay 

Membrane-bound α-lfucosidase activity was determined by using 400 μL of the resuspended cell pellet and 1600 μL of 3.5 mM *p*NP-Fuc in 100 mM phosphate buffer (pH 7). The mixture was incubated for 2 h at 37 °C; 200 μL aliquots of samples were removed every 30 min and centrifuged at 3110× *g* for 20 min at 4 °C to eliminate cells. The amount of *p*NP released was quantified in a spectrophotometer (Shimadzu UV-160A, Tokyo, Japan) at 410 nm. Activity from released α-lfucosidase was determined by using 200 μL of the supernatant and 800 μL of 3.5 mM *p*NP-Fuc in 100 mM phosphate buffer (pH 7). The mixture was incubated for 10 min at 37 °C, and the *p*NP released was recorded in a spectrophotometer at 410 nm. The molar extinction coefficient (ε) of the *p*NP under the conditions described was 7.8 mM^−1^ cm^−1^.

One unit of α-lfucosidase was defined as the amount of enzyme required to release 1 nmol of *p*NP per minute at pH 7 and 37 °C. Specific α-lfucosidase activity was defined as enzymatic activity (U) per mg of biomass, which was measured using a standard curve of dry weight cells.

### 3.6. Synthesis of Fucosyl-Oligosaccharides

The synthesis of fucosyl-oligosaccharides with α-lfucosidase from *L. rhamnosus* GG was performed using D-lactose, D-lactulose, and D-galactose as acceptor substrate at 200 mg/mL, and *p*NP-Fuc at 1 mg/mL as a donor substrate. The enzymatic transfucosylation reaction was accomplished in a stirred vessel in a total volume of 10 mL. Acceptor and donor substrates were dissolved in 100 mM phosphate buffer (pH 7), and 3.8 U/mL of α-lfucosidase was added. The mixture was incubated at 37 °C for 12 h, and aliquots were taken at regular intervals. Reaction was stopped by heating at 100 °C for 5 min. The carbohydrates of the mixture were analyzed as described below. Yields of transfucosylation were calculated based on donor substrate. The concentration of synthesized fucosyl-oligosaccharides was determined from the integration of area on HPLC chromatograms and the interpolation of peak area on a calibration curve of an external standard (2′-fucosyllactose). 

### 3.7. Composition of Synthesized Fucosyl-Oligosaccharide

Synthesized oligosaccharide was recovered by HPLC from a reaction using only lactose as an acceptor substrate. Collected samples were concentrated at 70 °C, and the purity of the compound was verified by HPLC (See Section 3.8).

The purified fucosyl-oligosaccharide was also subjected to mass analysis according to Guzmán-Rodríguez et al. [21]. For this purpose, the compound was concentrated to 1 mg/mL and analyzed using a Bruker Autoflex Speed MALDI-TOF/TOF system with a 1000 Hz Smart beam II laser. 2,5-Dihydroxybenzoic acid (DHB) was used as a matrix (5 mg/100 mL in 50% ACN/H_2_O), and 0.01 M NaCl was added as a cation dopant to increase signal sensitivity. The sample was spotted on a stainless-steel target plate, followed by the NaCl dopant and the matrix. The spot was dried in a vacuum prior to mass spectrometric analysis. MALDI-TOF MS via collision-induced dissociation (CID) was then performed to confirm the structure of the oligosaccharide. Tandem mass (MS/MS) spectrum was gained at 1 k eV collision energy with argon gas. 

To identify the monosaccharide moieties present in the structure of the oligosaccharide, an acid hydrolysis of the purified fucosyl-oligosaccharide was performed by adding 2 M HCl and incubated at 90 °C for 3 h. The hydrolysate was analyzed by HPLC to identify released monosaccharides (see Section 3.8).

### 3.8. Carbohydrate Quantification

Carbohydrates were determined by HPLC (LabAlliance, State College, PA, USA) using a Rezex RHM 7.8 mm × 300 mm column (Phenomenex, Torrance, CA, USA) for monosaccharides and an evaporative-light-scattering detector (ELSD) (Polymer Laboratories, Amherst, MA, USA). Samples were eluted with deionized water at a flow rate of 0.3 mL/min. Column temperature was maintained at 75 °C, and detector temperature was kept at 110 °C. A standard 2′-fucosyllactose curve was used to quantify the fucosyl-oligosaccharide. Concentrations of monosaccharides (d-glucose, d-galactose, and l-fucose) were calculated using standard curves for each one.

## 4. Conclusions

In this work, we report the release of α-lfucosidase from *L. rhamnosus* GG and its further application to synthesize fucosyllactose through transfucosylation reaction, which proved to be an alternative route for the production of fucosyl-oligosaccharide. Due to the fucosyl-oligosaccharide synthesized in this work having a composition similar to HMOs, they could provide the same advantages to human health as prebiotics and anti-infective compounds, which could of benefit to infants who are unable to be breastfed. 

## Figures and Tables

**Figure 1 molecules-24-02402-f001:**
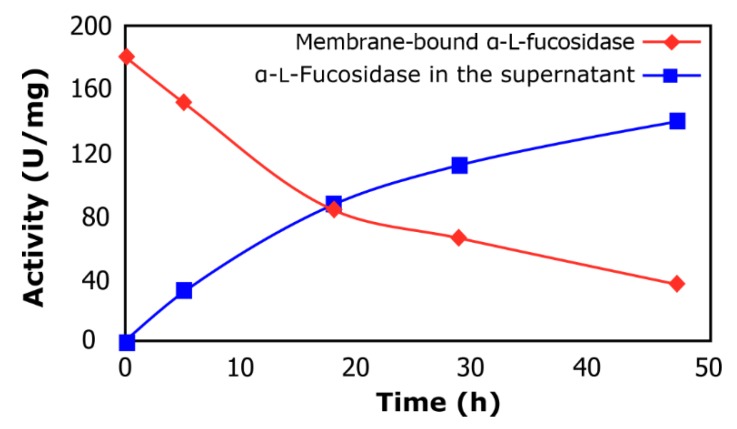
α-l-Fucosidase release from cells of *L. rhamnosus* GG. Membrane-bound α-lfucosidase (rhombus); α-lfucosidase in the supernatant (squares).

**Figure 2 molecules-24-02402-f002:**
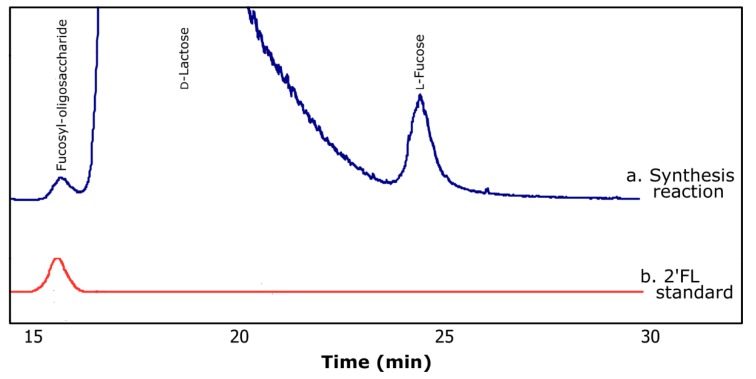
Typical chromatograms obtained by HPLC using ELSD for (**a**) transfucosylation reaction and (**b**) 2′-fucosyllactose standard (2′FL).

**Figure 3 molecules-24-02402-f003:**
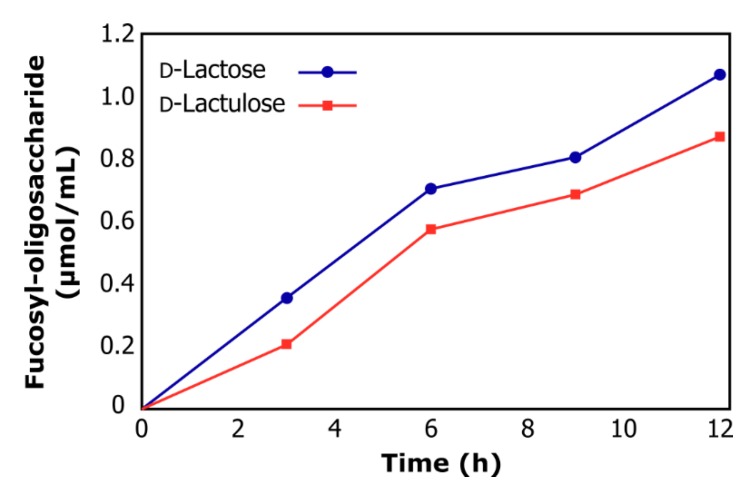
Reaction course of transfucosylation catalyzed by α-lfucosidase from *L. rhamnosus* GG using d-lactose (circles) and d-lactulose (squares) as acceptor substrates.

**Figure 4 molecules-24-02402-f004:**
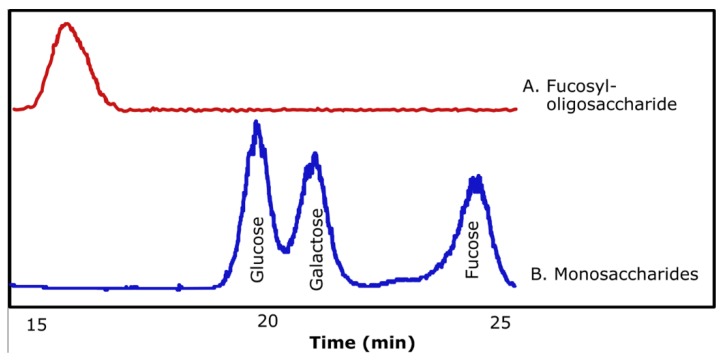
Chromatograms obtained by HPLC using ELSD of purified fucosyl-oligosaccharide (**A**) and monosaccharides obtained after acid hydrolysis of purified fucosyl-oligosaccharide (**B**).

**Figure 5 molecules-24-02402-f005:**
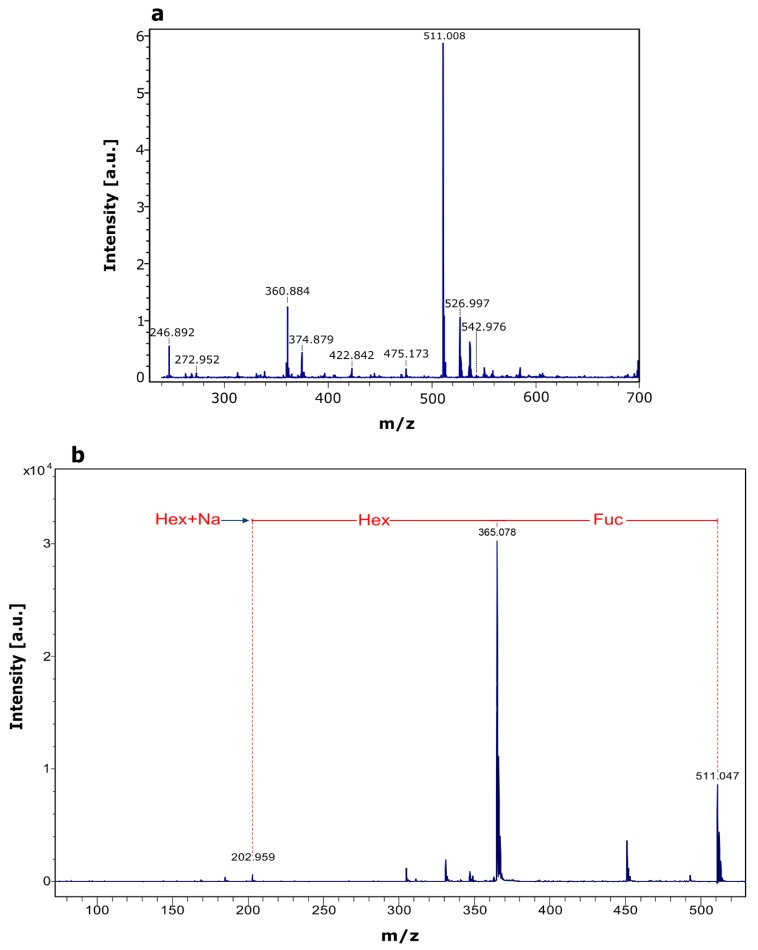
MALDI-TOF MS analysis of purified fucosyl-oligosaccharide. (**a**) MS spectrum of purified compound, (**b**) MS/MS spectrum of precursor ion (*m*/*z* 511.008).

**Table 1 molecules-24-02402-t001:** Synthesis of fucosyl-oligosaccharide using α-lfucosidase released from *L. rhamnosus* GG.

Acceptor Substrate	Oligosaccharide (µmol/mL)	Yield * (%)
d-Lactose	0.75	21
d-Lactulose	1.16	25
d-Galactose	0	0

* Yield was calculated based on the donor substrate.

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
