# Peer review of "Synthesis of Fucosyl-Oligosaccharides Using α-l-Fucosidase from Lactobacillus rhamnosus GG"

_molecules, 2019, doi:10.3390/molecules24132402_

Round 1
Reviewer 1 Report
In this manuscript, Escamilla-Lozano and coworkers describe the chemo-enzymatic synthesis of fucosyl glycosides using pNP-fucopyranoside and lactose/lactulose under the action of a fucosidase from Lactobacillus rhamnosus GG. The project describes a procedure to release the glycosidase from Lactobacillus and then utilize the solubilized preparation for attaching a fucose onto both lactose and lactulose. The results are sound and the organization of the manuscript makes them clear. There are a few questions/suggestions below, however, that may help to increase the impact of the paper.
1. How efficient is the fucosidase enzyme prep described here in comparison to others? The authors report percent yield, but it should maybe be corrected/normalized for the overall activity of the enzyme. Can that comparison be made?
2. The article is clear to state that the yield is based on the fucoside donor. That makes sense. The experimental should state how that was done procedurally. Was it based on an isolated mass of the glycoside product? Or calibrated HPLC integration? What?
3. Regioselectivity of the fucose transfer. Can the MS-MS give information on regioselectivty? It seems logical that there is only one regio-isomer, but the selectivity for the fucosidase is not discussed.
4. Minor items:
a. ELSD should be noted as the detection method in Figures 2 and 4.
b. FUCOs is a descriptive acryonym, but it may not be the best based on it’s similarity to the English curse word.
c. page 2 “potentially” is mis-spelled
d. Reference to “harmful chemicals” is misleading and should be changed. “potentially toxic” might be a safer way to convey the same message.
Author Response
In this manuscript, Escamilla-Lozano and coworkers describe the chemo-enzymatic synthesis of fucosyl glycosides using pNP-fucopyranoside and lactose/lactulose under the action of a fucosidase from Lactobacillus rhamnosus GG. The project describes a procedure to release the glycosidase from Lactobacillus and then utilize the solubilized preparation for attaching a fucose onto both lactose and lactulose. The results are sound and the organization of the manuscript makes them clear. There are a few questions/suggestions below, however, that may help to increase the impact of the paper.
How efficient is the fucosidase enzyme prep described here in comparison to others? The authors report percent yield, but it should maybe be corrected/normalized for the overall activity of the enzyme. Can that comparison be made?
Response: Purification information of other enzymes was added to compare with that obtained in this investigation (line73-76)
The sentence was modified (line 60)
2. The article is clear to state that the yield is based on the fucoside donor. That makes sense. The experimental should state how that was done procedurally. Was it based on an isolated mass of the glycoside product? Or calibrated HPLC integration? What?
Response: In the methodology section we describe how the yield was calculated (lines 216-218)
3. Regioselectivity of the fucose transfer. Can the MS-MS give information on regioselectivty? It seems logical that there is only one regio-isomer, but the selectivity for the fucosidase is not discussed.
Response: Matrix assisted laser desorption ionization-time-of-flight-mass spectrometry (MALDI-TOF-MS) analysis may now generate fast profiles but does not allow isomer separation.
A paragraph was added to discuss the regioselectivity of the fucosidases (Line 96-114)
4. Minor items:
a. ELSD should be noted as the detection method in Figures 2 and 4.
Response: The description of figures 2 and 4 was corrected (lines 88-89, 145-146)
b. FUCOs is a descriptive acryonym, but it may not be the best based on it’s similarity to the English curse word.
Response: The acryonym was eliminated
c. page 2 “potentially” is mis-spelled.
Response: The word was corrected (line 65)
d. Reference to “harmful chemicals” is misleading and should be changed. “potentially toxic” might be a safer way to convey the same message.
Response: The sentence was corrected (line 39)
Reviewer 2 Report
The manuscript by Cruz-Guerrero et al. describes the isolation of a alpha-L-fucosidase from a lactobacillus and its use for the enzymatic synthesis of 2’-fucosyllactose from p-nitrophenyl-alpha-L-fucopyranoside and lactose. The manuscript is clearly written and the performed experiments are precisely described.However, it remains unclear how the structure of the obtained trisaccharide is finally proven. The authors only compare the HPLC retention times of their product with an authentic sample of 2’-fucosyllactose although the peaks appear to relatively broad. The presented MALDI-TOF spectra of the purified compound is also no unambiguous proof for the structure. 3-Fucosyllacctose which might also be a product of the fucosidase reaction cannot completely ruled out by MS. The authors should add a GC-MS analysis which had previously been shown to clearly distinguish between 2’-fucosyllactose and 3-fucosyllactose (see: R. Balogh et at., J. Pharmaceut. Biomed. 2015, 115, 450-456). Alternatively, NMR spectra of the isolated compound whould also unambiguously prove its structure. It is also unclear how exactly the yield for the fucosylation was calculated (by weighing of the isolated product or by measuring the peak areas in the HPLC chromatogram). In the latter case (yield from chromatogram) the authors should state whether or not a calibration of the chromatogram with an authentic sample of 2’-fucolyllactose had been done. In addition, the following corrections should be made: a) line 96; methyl-ß-D-galactopyranoside instead of methyl-ß-D-galactose; b) line 126; fucose is also a hexose.
Therefore, three hexoses awere formed during hydrolysis; c) line 155; explain what “isotonic solution” means
Author Response
The manuscript by Cruz-Guerrero et al. describes the isolation of a alpha-L-fucosidase from a lactobacillus and its use for the enzymatic synthesis of 2’-fucosyllactose from p-nitrophenyl-alpha-L-fucopyranoside and lactose. The manuscript is clearly written and the performed experiments are precisely described.
However, it remains unclear how the structure of the obtained trisaccharide is finally proven. The authors only compare the HPLC retention times of their product with an authentic sample of 2’-fucosyllactose although the peaks appear to relatively broad. The presented MALDI-TOF spectra of the purified compound is also no unambiguous proof for the structure. 3-Fucosyllacctose which might also be a product of the fucosidase reaction cannot completely ruled out by MS. The authors should add a GC-MS analysis which had previously been shown to clearly distinguish between 2’-fucosyllactose and 3-fucosyllactose (see: R. Balogh et at., J. Pharmaceut. Biomed. 2015, 115, 450-456). Alternatively, NMR spectra of the isolated compound whould also unambiguously prove its structure. It is also unclear how exactly the yield for the fucosylation was calculated (by weighing of the isolated product or by measuring the peak areas in the HPLC chromatogram). In the latter case (yield from chromatogram) the authors should state whether or not a calibration of the chromatogram with an authentic sample of 2’-fucolyllactose had been done.
Response: The composition of the oligosaccharide synthesized was verified with the experiments carried out, since both, the acid hydrolysis/HPLC and MALDI-TOF-MS-MS allowed the identification three hexoses and one of them was fucose, with which it can be said that fucosyllactose was synthesized. The methodology suggested by the reviewer cannot be done because the institution does not have GC-MS. However, we consider that our results are relevant given that the fucosyllactose synthesis is verified, and according to the reported by other researchers it is possible that it has some biological activity (lines 157-158) since both the 2-fucosyllactose, 3-fucosyllactose or 6-fucosyllactose isomer has biological activity. In future studies, the identification of the compound by NMR will begin.
Regarding the peaks obtained in the chromatograms, the retention times of the standard and the synthesized oligosaccharide were included in the report, as can be seen, they are very similar, so we can rely on the identification of the compound (lines 82, 83). Moreover, in the methodology section we describe how the yield was calculated (lines 216-218)
In addition, the following corrections should be made:
line 96; methyl-ß-D-galactopyranoside instead of methyl-ß-D-galactose;
Response: The word was corrected (line 121)
b) line 126; fucose is also a hexose. Therefore, three hexoses awere formed during hydrolysis;
Response: The sentence was modified (lines 151-152)
c) line 155; explain what “isotonic solution” means
Response: The solution was described (line 191)
Round 2
Reviewer 2 Report
After incorporation of the recommended corrections the paper should be published in Molecules. It would have been a quite excellent paper would the unambiguous structure assignment had been made. The referee hopes that the authors will do that in the future as announced though.